# Defining Fragmentation Patterns of Archaeological Bone Remains without Typologies: A Landmark-Based Approach on Rodent Mandibula

**Marine Durocher** [1,2,3,*] , **Sandrine Grouard** [1] , **Violaine Nicolas** [2] , **Renan Maestri** [4] **and Allowen Evin** [3]

1   Laboratoire Archéozoologie, Archéobotanique: Sociétés, Pratiques, Environnements,
    Muséum National d'Histoire Naturelle, CNRS, 75005 Paris, France; sandrine.grouard@mnhn.fr
2   Institut de Systématique, Évolution, Biodiversité, Muséum National d'Histoire Naturelle, CNRS,
    Sorbonne Université, EPHE, Université des Antilles, 75005 Paris, France; violaine.colin@mnhn.fr
3   Institut des Sciences de l'Evolution-Montpellier, UMR 5554-ISEM, CNRS, Université de Montpellier, IRD,
    EPHE, 34095 Montpellier, France; allowen.evin@umontpellier.fr
4   Instituto de Biociências, Universidade Federal do Rio Grande do Sul (UFRGS), Porto Alegre 90040-060, Brazil;
    renanmaestri@gmail.com
*   Correspondence: marine.durocher@mnhn.fr

**Abstract:** Fragmentation is a recurring feature of archaeological faunal material, and impacts many aspects of zooarchaeological studies from taxonomical identification to biometric studies. It can result from anthropic and natural actions that occurred respectively before and/or after bone deposit. While several bone fragmentation typologies have been described, they are currently based on both macroscopic observations and researcher subjectivity and lack the universality necessary for inter-study comparisons. To fulfill this need we present a standardized landmark-based protocol for the description and quantification of mandibular fragmentation patterns, using two insular rodents of different sizes as models. The rice rats (Oryzomyini tribe) and the agouti (*Dasyprocta*) from the Lesser Antilles were abundant during the pre-Columbian Ceramic Age (500 BCE-1500 CE). Their mandibles' shapes were quantified using the coordinates of 13 2D-landmarks. We show that landmark-based measurements can be used to:—assess the preservation differences between taxa of the same taxonomic group (e.g., rodents),—estimate the level of preservation of a skeletal part (e.g., mandible),—describe fragmentation patterns without pre-existing typologies and—facilitate the application of geometric morphometric methods to fragmented archaeological material. Our novel approach, leveraging fragmentation analyses and establishing specific fragmentation patterns, frees itself from existing typologies and could be systematically applied to future research.

**Keywords:** geometric morphometrics; taphonomy; zooarchaeology; mandible; fragmentation; rodents

## 1. Introduction

Archaeological bones are studied to investigate the relationships between past human societies and animals as well as the environment. Bone fragmentation results from pre- and/or post-depositional taphonomic processes that may be caused by natural or anthropogenic actions [1,2]: butchering and hammering of bones, trampling, climato-edaphic effects, non-human biological agents (e.g., carnivores), and breakage resulting from excavation, transportation, and storage [3]. One common aim of taphonomic studies is to characterize fragmentation patterns throughout the assemblage in order to compare and recognize the taphonomic processes involved [4–12].

Alteration of faunal remains affects preferentially specific bones and specific areas on the bones [13]. Compared to other bones, mandibles of large [14] and small mammals [15] tend to be well preserved in archaeological deposits as a result of their high structural density. However, the bone density differs between the taxa and the bone mineral composition varies according to the age of the animal, which induces different structural and mechanical

properties [16]. As a result, within the same group of taxa, e.g., rodents, mandibles do not always have the same mechanical properties [15]. Despite these observations, mandibles tend to show a non-random fragmentation and can be divided into two parts: the anterior part or *corpus* which bears the teeth, and the posterior part or *ramus* that is attached to the skull. The *corpus* is generally better preserved [15,17]. However, the ability to identify to which species or group of species belongs a mandibular fragment depends on its size and the anatomical part represented [18,19].

Landmark-based approaches, such as the ones commonly used in geometric morphometrics [20–22] employ two or three-dimensional coordinates of points to quantify the shape and the size of objects. Many morphometrical and statistical analyses do not accept missing data and require complete specimens. In the case of fragmented material—a common case in archaeology or paleontology—one frequently reduces either the number of specimens [23] or the number of variables [24]. Doing so, a trade-off has to be found between the number of available specimens and the number of available landmarks. Morphometric-based techniques for fossil reconstruction offer the possibility to estimate the location of missing landmarks in order to maximize the number of specimens studied [23,25]. However, such techniques require specimens that can be used as reference—a challenge when the number of available specimens per archaeological site or period is limited and with each specimen displaying only subtle morphometric differences. An alternative approach to studying fragmented shrew mandibles was suggested by Cornette and collaborators (2015) [26]. This approach is based on coordinates of points along the external outline (sliding semi-landmarks) of the mandible from a 2D image of its lateral labial view. Analyses are then performed, without estimating missing parts, but using a pre-established fragmentation typology describing seven fragments ranging from the complete mandible to the single ascending *ramus* [26].

On another hand, other typologies of fragmentation patterns have been described for the mandible [18,27,28] but they vary from one study to the other, relying on the taxa and geographical areas. In this paper, we propose a new, objective, and quantitative method for the analysis of mandibular fragments. With this approach, no zonation typology is required and fragments of all sizes can be studied as long as at least the coordinates of two landmarks can be obtained. This method will also facilitate the selection of the largest number of fragments that can be used in future geometric morphometric studies while retaining the capacity to characterize shape variations, applied here in a case study on modern oryzomyine rodents.

## 2. Materials

Two rodents were used as a model: the rice rat (Muroidea, Cricetidae, Oryzomyini tribe) and the agouti (Caviomorpha, Dasyproctidae, *Dasyprocta* spp.). These rodents are recovered from Pre-Columbian archaeological sites of the Lesser Antilles. Both rodents were actively consumed by pre-Columbian human populations as evidenced by the presence of butchery cuts and burn marks [29–33].

We analyzed a total of 833 archaeological mandibular fragments among which 767 belong to rice rats and 66 to agoutis. Specimens originate from 17 Lesser Antilleans islands and 45 archaeological sites (Table S1.1). The mandibular fragments studied originate from the same archaeological contexts as the teeth studied elsewhere [34]. Specimens are housed in the collections of the Regional Service of Archaeology (Service Régional de l'Archéologie, Direction des Affaires Culturelles (DAC)) of Guadeloupe and Martinique, at the Muséum national d'Histoire naturelle de Paris (MNHN, France), and at the Florida Museum of Natural History (FLMNH, USA). These remains date from the Ceramic Age (500 BCE–1500 CE) and were part of faunal assemblages excavated and studied heterogeneously. Three categories can be described: the first category (T-1) corresponds to the most recently excavated sites from the 2000–2010s, water sieved with two meshes (1 to 2.7 mm screen) and for which the faunal assemblages have already been studied zooarchaeologically. This category represents 73% of the Oryzomyini and 35% of the Dasyproctidae mandibles included in

the present study. The second category (T-2) includes sites from older excavations dated from the 60' and 90' where only the "almost-complete" bones were recovered (sometimes with sieving). This corresponds to 11% of the Oryzomyini and 25% of the Dasyproctidae studied. The third category (T-3) includes remains recovered in heavy boxes of sediments and mixed artifacts never properly studied. This last category represents 16% of the Oryzomyini and 40% of the Dasyproctidae. Note that the most fragmented mandibles might not have been recovered and included in our dataset because of the method of study and recovery of the material used for the T-2 and T-3 excavation sites. Conversely, the mandible remains that were recovered from the T-1 category have been studied exhaustively and likely include all types of fragments, due to the good conditions and time dedicated to these zooarchaeological studies.

In addition, 554 complete mandibles of modern rice rats originating from the South American continent (Table S1.2) serve as a basis for comparison in the direct application of the method developed. All these mandibles have already been analyzed [35]. These mandibles belong to 24 genera, are not fragmented, and therefore preserve all the morphological information associated with taxonomic differentiation.

## 3. Methods

The lateral view of each archaeological mandible fragment was photographed respecting the anatomical positioning of the mandibles. For agoutis, the correct position of the mandibles was checked using a bubble level. In the case of the smaller Oryzomyini mandibles, the position could only be visually inspected. Photographs of all the archeological material were captured by Marine Durocher (M.D.) using a CANON EOS 80D camera, while South American Oryzomyini was captured by Renan Maestri (R.M.) using a Nikon P100 camera [36]. Results of the two data sets were contrasted but not pooled due to methodological and technical interstudy differences (camera and mandible positioning differ). A maximum of thirteen 2D landmarks were digitized by M.D. on each fragment using the TPSdig2 software package [37]. A complete mandible was therefore characterized by 13 2D coordinates while a minimum of two coordinates were recorded on the smallest fragments. The protocol already established for South-American Sigmodontinae [35] was used and adapted to agouti. The protocols for the two rodents differ only by the position of landmark 13 which corresponds to the intersection between the *corpus* and the ventral insertion area of the anterior deep masseter in the agouti and to the most posterior-ventral limit of the mandibular symphysis for Oryzomyini (Table S2).

First, we assessed the number of landmarks present in each specimen (from 2 to 13). Frequency in the number of available landmarks was quantified, and visualized by a pie chart (package 'ggplot 2' v3.1.1; [38]). Then, the frequency of each landmark presence was quantified, and visualized with a bar plot (package graphics v3.5.3; [39]).

The second step was to define all possible fragmentation patterns, i.e., all the combinations of 2 to 13 landmarks according to the fragmentation degree of the mandible (e.g., landmarks #1, #4, and #12; landmarks #2 and #6; . . . ). For each number of landmarks (from 2 to 13) the frequencies of the three most common patterns were quantified and visualized with box plots. The R function "printbest" (available upon request) allows us to represent the three most frequent fragmentation patterns in the dataset per number of digitized landmarks. The difference of preservation between the two taxa was assessed with khi2 tests.

In order to assess whether variation in the way remains was recovered from the excavation affects the number of identified fragments and fragmentation patterns, we observed the representation frequency of the most common fragmentation pattern in the assemblage, according to the three types of contexts and for the two rodents.

Finally, in order to determine the characterization capacity of the shape variation of the patterns derived from the methods (second requirement of a geometric morphometrics study), the archaeological patterns, as well as the complete protocol (13 landmarks), were applied on oryzomyine specimens from reference collection. The number of principal

components (PCs) that maximize the cross-validation percentage (CVP) was selected. It allowed us to calculate a CVP common to all patterns as well as the complete protocol.

## 4. Results

### 4.1. Mandibular Preservation

Only 0.8% (N = 6/767) of the Oryzomyini and none of the agouti mandibles were complete and measured by the full set of landmarks (Figure 1). For Agoutis, a maximum of 10 landmarks could be measured but only on a single specimen (1.9%). In comparison, 10 landmarks could be measured on 6.3% (N = 48/767) of the Oryzomyini mandibles (Figure 1). The same contrast is observed for the smallest fragments with only 0.7% (N = 5/767) of the Oryzomyini against 6% (N = 4/66) of the Dasyproctidae mandibles that are represented by only 2 landmarks (Figure 1). For both taxa, the number of fragments varies depending on the number of landmarks available (Figure 2). Most measured mandibles allow the positioning of six landmarks for the Dasyproctidae (21.6%; N = 14/66) and seven for the Oryzomyini (24.5%; N = 188/767).

All landmarks are not equally preserved but their occurrence displays the same trend for the two taxa (Figure 2). For both taxa, the superior part of the *corpus* of the mandible (quantified by landmarks #1 to #5), as well as the lower part of the *corpus* (quantified by landmarks #12 and #13) are particularly well preserved compared to the *ramus* (landmarks #6 to #11).

For each digitized landmark, the distribution pattern is also quite similar between *Dasyprocta* and Oryzomyini but some variations can be detected. For instance, while landmarks #3 and #12 are the most often present for both species, landmark #9 is never measured on agouti but is present on 20% of the rice rat mandibles. In conclusion, even if the mandibles of the two rodents seem to have similar preservation, the size of the fragments differs between the two (i.e., number of landmarks, Figure 1) as well as the preserved anatomical parts (i.e., location of landmarks, Figure 2). Altogether these results show that mandible preservation deeply differ between taxa ($\chi^2 = 25.6$, df = 12, *p*-value = 0.012). As a consequence, the two taxa were analyzed separately in further analyses.

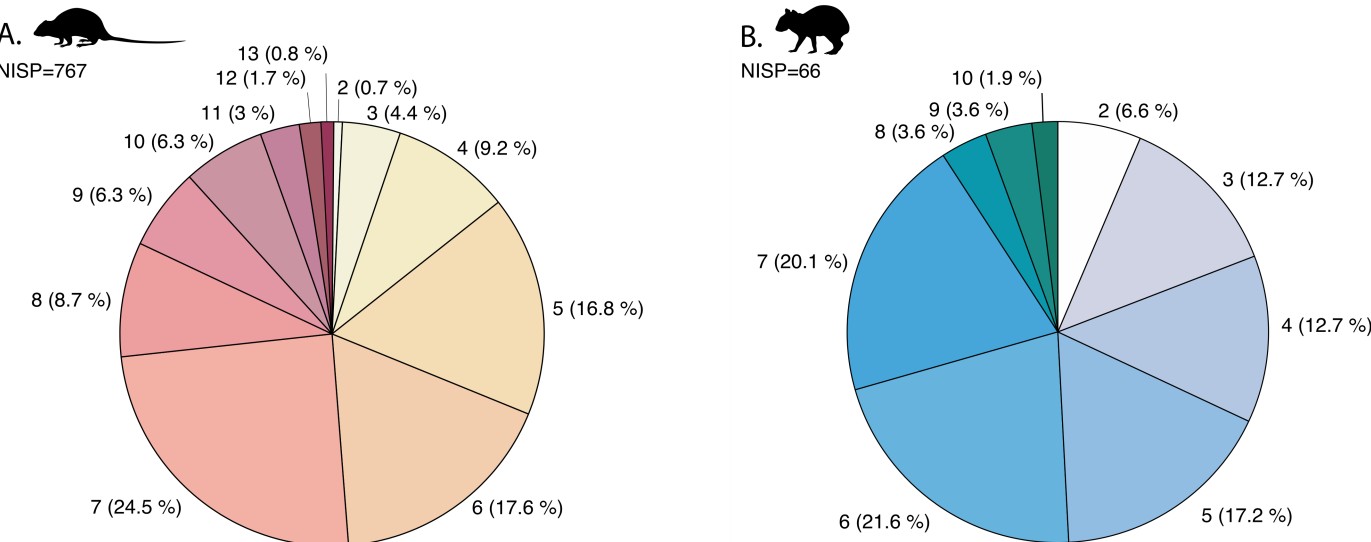

**Figure 1.** Pie chart representing the preservation for Oryzomyini (**A**) and *Dasyprocta* (**B**) mandibles according to the number of available landmarks (from 2 to 13). See Table S2 for the number of Oryzomyini and *Dasyprocta* specimens (NISP) for each digitalized landmark.

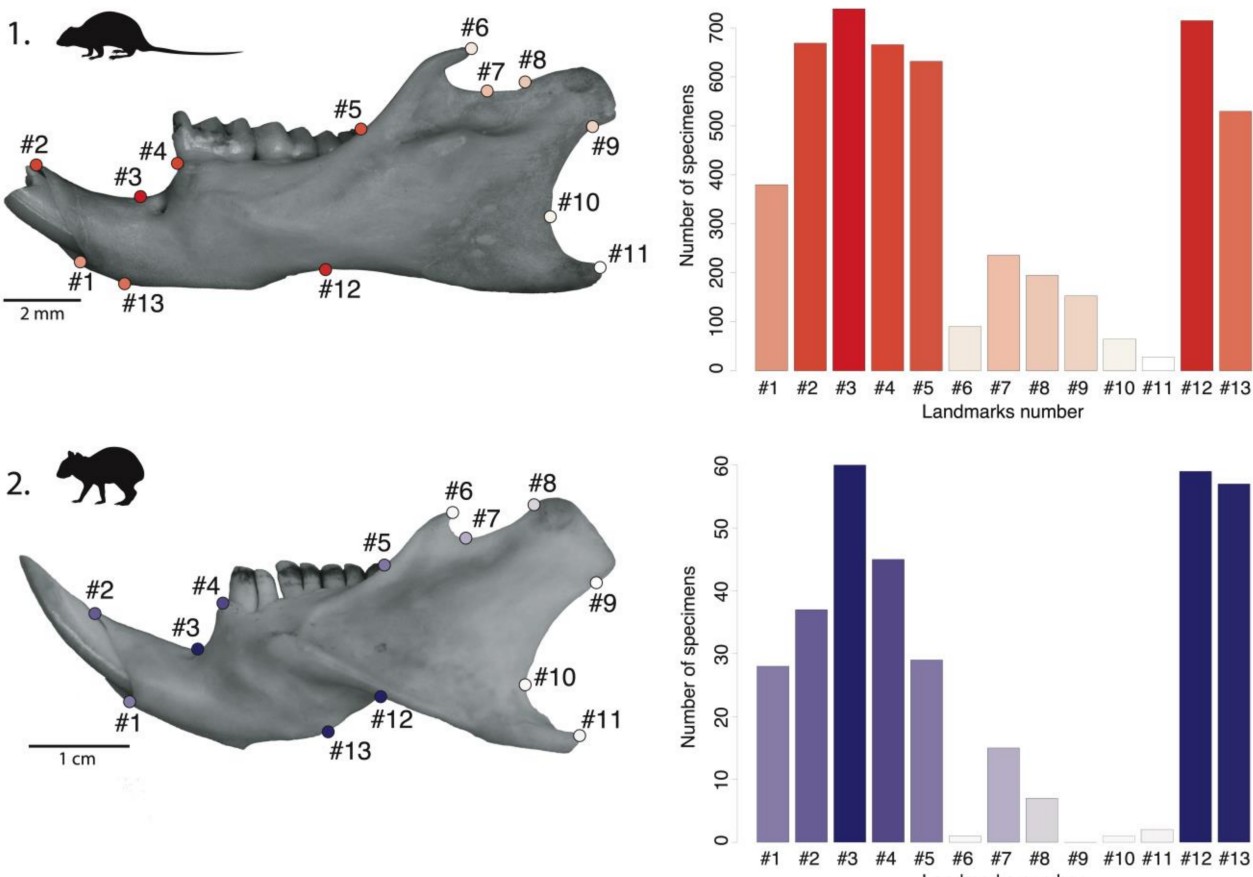

**Figure 2.** (**Left**) Protocols for positioning the landmarks for the Oryzomyini (**1**) and for the *Dasyprocta* (**2**) colored according to the values of the histogram (**Right**) representing the number of mandibles on which each type of landmark was measured (Oryzomyini NISP = 767, *Dasyprocta* NISP = 66). (See Table S3 for the landmark descriptions).

*4.2. Fragmentation Patterns*

For each number of measured landmarks (from 2 to 13 for rice rats and from 2 to 10 for agoutis) the three most frequent patterns were identified and visualized (categories P1 to P3, Figure 3). Not surprisingly, the smaller the number of landmarks measured the greater the number of fragments available. On the mandibular *corpus*, up to 7 landmarks can be digitized, which corresponds to the majority of quantified fragments.

The category P1 corresponds to the most represented pattern in the assemblage and allows from 2.09% (13 landmarks) to 89.96% (for 2 landmarks) of the oryzomyine mandibles to be measured. This category allows to measure up to 33% more mandibles than P2 and up to 38% more than P3. Within the Dasyproctidae, P1 allows us to measure from less than 1% (for 10 landmarks) to 98.21% (for 2 landmarks) of the assemblage, which is up to 38% more fragments than P2 and P3 (Figure 3).

These patterns can help to define which anatomical parts of the mandible could be studied with the maximum number of specimens. They are not used here to establish a typology. However, the 11 most present patterns (P1) of the oryzomyine depict an anatomical evolution along the mandible (Table S4), starting from its anterior to its posterior part, in accordance with the morphology and the mechanical properties of the mandible (Figure 3).

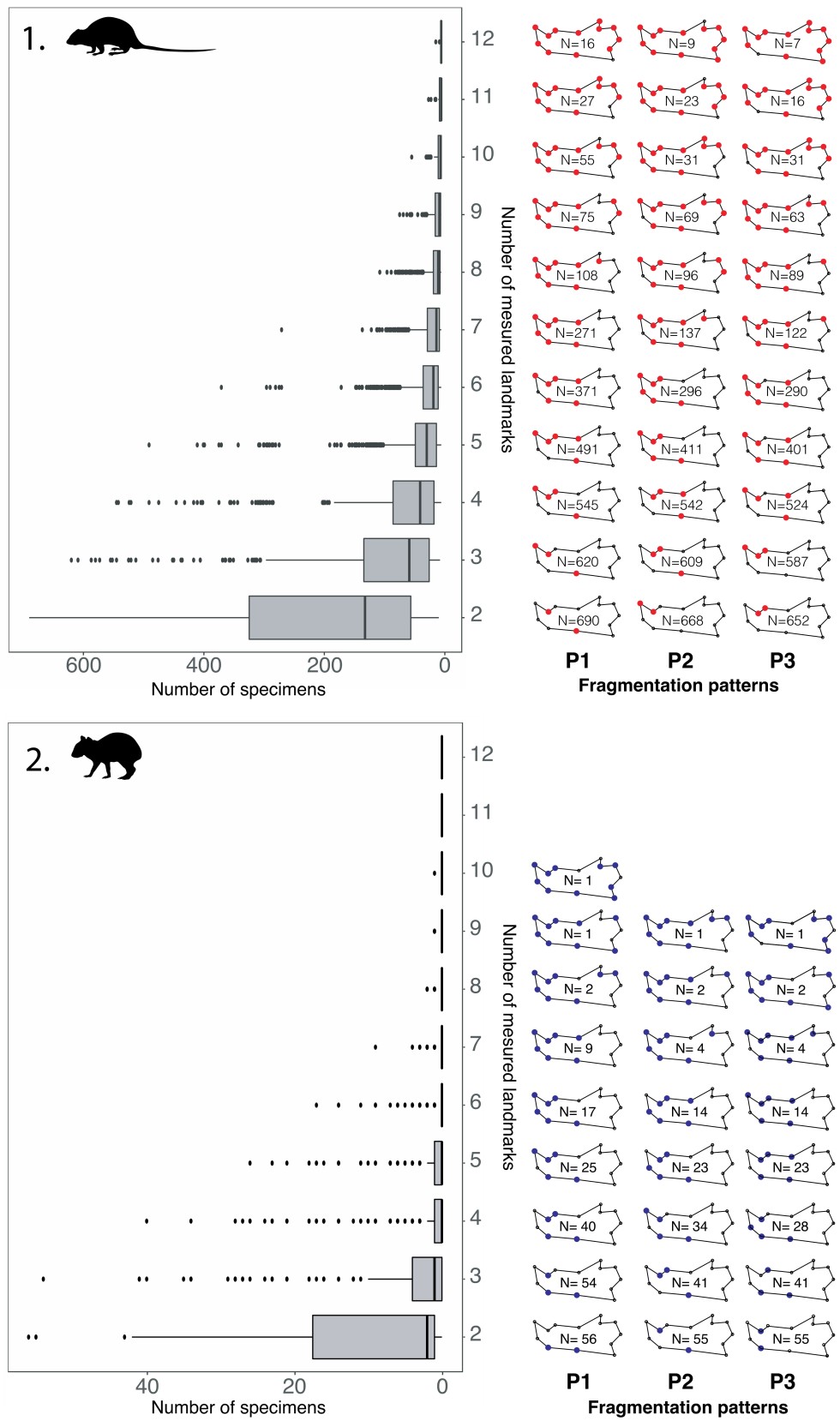

**Figure 3.** (**Left**) Number of fragmentation patterns for N landmarks on Oryzomyini (**1**) and *Dasyprocta* (**2**) mandibles, (**Right**) categories of the first three most frequent fragmentation patterns for N landmarks (P1, P2, P3).

Because the excavation techniques can have an impact on finding a fragment, the three types of recovery were compared (Figure 4). For both taxa, the three types show the same proportions of remains independently of the size of the fragment (Figure 4). However, a much larger number of fragments come from T-1 suggesting that the recovery methodologies may influence the number but not the size of the fragments identified.

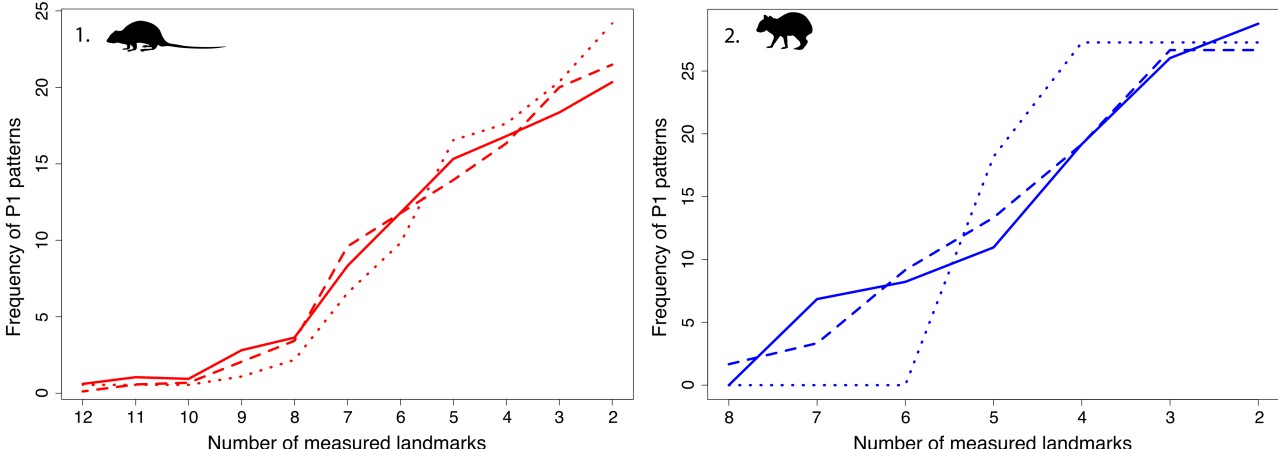

**Figure 4.** Frequency of the most frequent fragmentation patterns (category P1) in the assemblage, according to the three types of recovery and study of the faunal material (T-1: solid line, T-2: dashed line, T-3: dotted line), for (**1**) Oryzomyini mandibles and (**2**) *Dasyprocta* mandibles.

### *4.3. Which Pattern to Select for a Geometric Morphometrics Study? The Case of Oryzomyini*

When dealing with a geometric morphometrics study numerous criteria must be considered to select the most suitable mandibular fragmentation pattern(s). The largest number of individuals should be targeted. Yet, the patterns should maintain the ability to discriminate groups.

In order to assess the taxonomic information carried by the various patterns, the efficiency to discriminate the Oryzomyini genera was assessed for the most frequent patterns (Figure 3, Table S4). The discriminatory power of the complete mandibula was confronted with the patterns predominantly found in the archaeological record.

With the studied case of Oryzomyini, in order to determine the fragmentation pattern(s) that best meet these two criteria we used a reference sample comprising 554 complete continental oryzomyine mandibles. For each mandible, the taxonomic attribution was used at the genus level to consider the maximal number of individuals. This case study allows us to test the capacity of the different fragmentation patterns to characterize the phenotypic variation between genera.

When studying the 767 archaeological mandibular fragments of Oryzomyini, 27 most frequent fragment patterns are defined according to three counts: the category of frequency of measurement (P1, P2, P3), the number of landmarks measured (between 4 and 12), and the percentage of the archaeological *corpus* that could be measured with each of these patterns (Table S5). This selection meets an important requirement of a geometric morphometrics study, namely the search for the largest number of individuals studied.

To determine the discriminant power of the various fragmentation patterns, the 27 most common fragments in the archaeological records (3 most frequent patterns for 4 to 12 landmarks) were compared to the complete mandible (i.e., including 13 landmarks). The cross-validation percentages decrease with the number of landmarks used to characterize the mandible (Figure 5). While the complete protocol succeeded in correctly identifying at least 79% of the specimens, the protocols including 4 landmarks correctly identify ~49% of the specimens. In all those comparisons the random correct cross-validation turns around 4% (i.e., 100/24 genus). The inter-continental genus cross-validation percentages were calculated (Figure 5) and three patterns provided as good discrimination as the complete protocol. Those protocols are: P1-7LM (size: CVP = 23.3% (CI:19.5–26.8%); shape: 52.3%

(CI:47–57.6%)), P2-6LM (size: 23.3% (CI:19.5–27.6%); shape: 48% (CI:44.2–52. 2%)) and P3-6LM (size: 23.5% (CI: 20.7–27.2%); shape: 44.6% (CI: 39.8–48.9%)). These three patterns, which include 6 or 7 landmarks, therefore carry a strong taxonomic signal and can be applied to the archaeological assemblage.

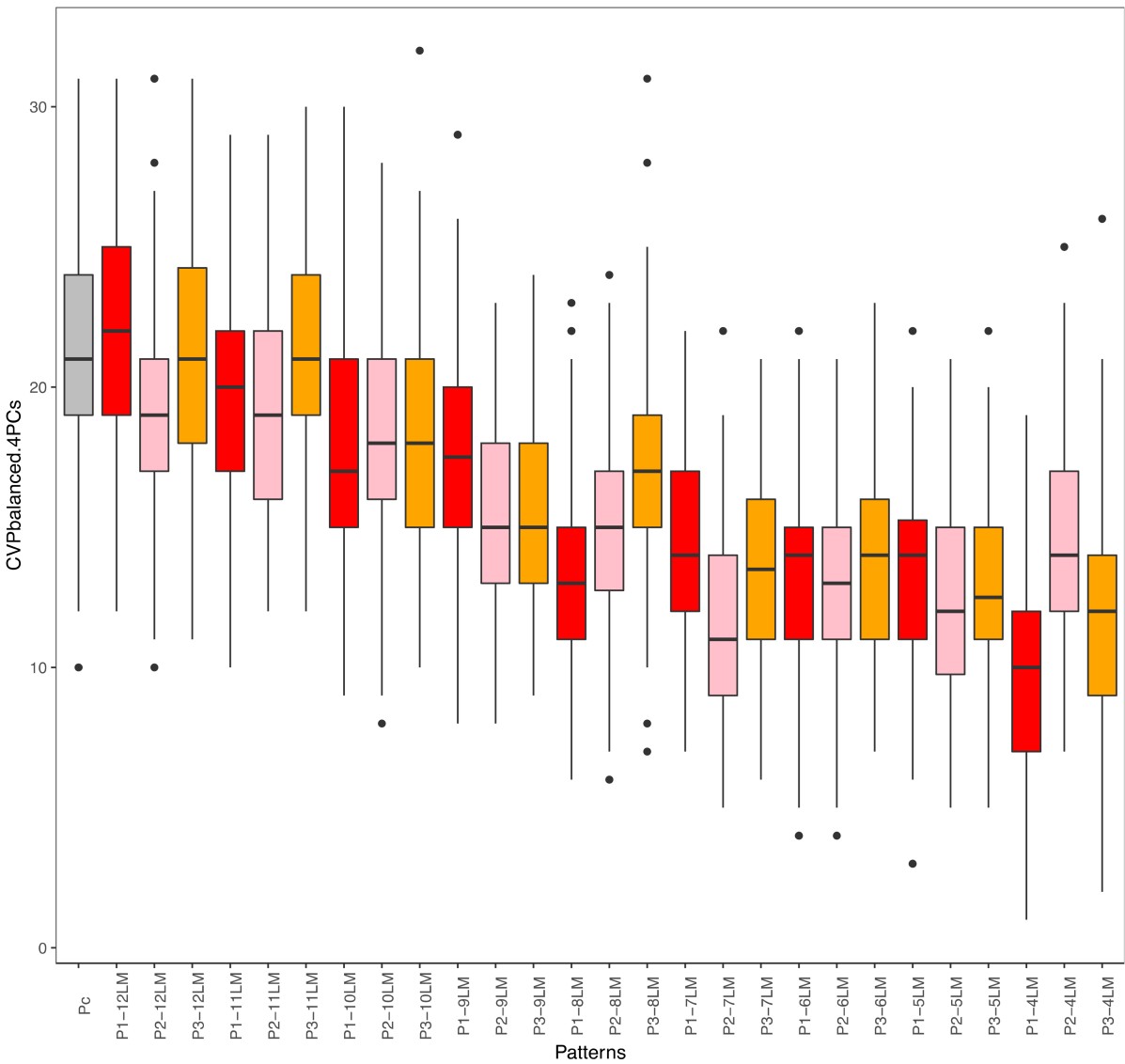

**Figure 5.** Boxplot representing the distribution of cross-validation percentages (CVP) for 4 PCs of each pattern (between 4 and 12 landmarks), according to the genus of attribution of specimens from the reference collections. Grey: Pc: complete protocol, Red: P1: category of the first most frequent fragmentation patterns with N landmarks, Pink: P2: second category, Orange: P3: third category.

## 5. Discussion

### 5.1. Impact of Mandibular Properties on Preservation

Our analyses revealed better preservation of the anterior part of the mandible for both taxa which likely results from the mechanical properties of the mandibles leading to better preservation of the *corpus* than the *ramus* [15,17]. Located on the posterior part of the mandible, the *ramus* has a complex shape with three processes (coronoid, condyloid, and angular), and a large flat and thin area, the masseteric fossa, more frequently fragmented. The angular process is thin and elongated, more developed in agoutis than rice rats, but it never exceeds the condyloid process. The condyloid process is developed on the top of the collar. It is wide and thicker in its rostral part. The coronoid process is thin, elongated, and

curved over the mandibular incision. For the rice rat, the coronoid process is higher than the condyloid process, while it is smaller for the agouti (anatomical descriptions adapted to our taxa [40,41]). These three processes are extended beyond the main shape of the mandible and are fragility points that could explain the poor preservation of the *ramus*. The two most frequently measured landmark classes on the mandibles of both Oryzomyini and *Dasyprocta* (Figure 2) are representative of these mechanical properties, with the anterior part of the mandible (*corpus*) having much better preservation than the *ramus*. The degree of preservation of the different parts of the mandible can also be quantified. For example, in archaeological rice rats, the degree of preservation of the coronoid process is much lower than that of the incisor canal (12% versus 76%).

*5.2. Application to Zooarchaeological Studies*

In the patterns defined here by a methodology not based on diagnostic zones typology, the anterior part of the mandible is well represented. If the mandible is fractured in two pieces, between the *corpus* and the *ramus*, one should expect to identify these two parts in similar proportions, which is not the case here. Two factors may explain this under-representation of the posterior part. Firstly, the different shapes of the two parts of the mandible must be considered. As described before, the *ramus* is composed of three processes that contain significant weakness points that will lead to its fragmentation. Indeed, the different patterns obtained clearly showed that the first breaking parts in the most frequent patterns of fragmentation concern these processes. Moreover, in the entire area of the masseteric fossa, the bone thickness is reduced and the bone fragile. On the other hand, the shape of the *corpus* is less complex and tends to maintain its integrity, despite the taphonomy. The second factor is related to the identification of the fragments themselves. As the posterior part of the mandible tends to be more fragmented than the anterior part, the fragments will be both smaller in size and more variable in shape. These fragments will therefore tend to be classified as "unidentified remains" during the zooarchaeological study since these tiny fragments do not allow a taxonomic attribution or even—sometimes—a complete anatomical identification.

Despite these difficulties, the excavation and study methods used to recover the fragments do not seem to have impacted the representativeness of material fragmentation, at least in the case of these rodent mandibles (Figure 4). However, it cannot be excluded that the under-representation of mandibles from archaeological sites (T-2 & 3, both less precisely studied) in our assemblage are symptomatic of the difference in the recovery process of the remains. For simplicity only the three most frequent patterns were presented (Figure 3), but the proposed methodology enables targeting other specific patterns which may not have been represented here. An overview of all existing patterns could allow us to further develop the analysis. For example, differential preservation through time or space could be explored. It might also be possible to associate fragmentation patterns with more specific taphonomic agents and then quantify their ratio over the fragmentation of the archaeological faunal assemblage.

*5.3. Interpretation of Differential Preservation between Taxa*

The comparison of two taxa belonging to the same taxonomic class, present in the same archaeological deposits, helps to better understand differential preservation. Direct observation shows that agouti mandibles were more fragile than those of the rice rats. Despite the fact that the agouti mandibles are larger, the bone is proportionally thinner and therefore more subject to fracture. With the approach using no *a priori* typology, we were able to establish that the mandibular preservation differs between the two taxa. We suspect that these differences are due to the density of the mandible, and the size of the alveolar tooth row, but this should be further analyzed and quantified in order to better understand the preservation biases that exist within a faunal assemblage. Agoutis are under-represented compared to rice rats in archaeological sites. Classically, this would be interpreted as a lower proportion of agoutis in the diet of pre-Columbian populations or

as a differential anthropological treatment (butchery, cooking, taming practices, or bone industry) of the two rodents as it has been described for other taxa [42]. Indeed, these remains come from human occupation sites, and butchery cuts and burn marks present on the bones suggest an anthropic accumulation. However, we cannot exclude that their under-representation could also be, at least partially, linked to poorer preservation of the bone remains of this taxon.

*5.4. Future Geometric Morphometrics Application*

Finally, this landmark-based approach should simplify the application of geometric morphometrics to fragmented archaeological material (see Section 4.3). Indeed, the use of geometric morphometrics is increasing in zooarchaeological studies. It is used, for example, to determine taxa that cannot be distinguished by classical comparative anatomy [43,44], or to characterize shape and size variations induced by domestication and/or captivity [45–47]. Derivatives of geometric morphometrics, through landmark-based approaches, like the one proposed in this article, are also in development [48]. They offer more objective and quantitative approaches than traditional methodologies.

Geometric morphometrics often requires the use of complete specimens and therefore the landmark coordinate acquisition protocols require to be suitable for characterizing—anatomically—meaningful shape variations while maintaining the largest number of individuals. The method we propose enables researchers to estimate and describe fragmentation without *a priori* and to select among all the existing patterns the most common ones that can be used for further analyses.

The objective and quantitative definition of patterns that can be established by this method also allow us to avoid the usual issues related to the typologies: i.e., the fragmentation may vary depending on the sites studied, their condition of preservation, and the taxa studied. Finally, the use of the exact same protocol to assess the typology for each of the bones will limit the bias induced by the subjectivity of discrete characters.

## 6. Conclusions

The proposed landmark-based approach enables us to identify the number and location of the landmarks to be used to increase the number of archaeological fragments suitable for a geometric morphometric study. Archaeological material—which by definition has spent several hundred (or thousands) years buried—is frequently subject to fragmentation. This was reflected in the mandibles available for geometric morphometric analysis. With this new method, all sizes and types of fragments can be analyzed, making it easier to obtain a broad overview of the faunal assemblage diversity. So far, most taphonomic and zooarchaeological (fragmentation patterns) studies are both based on discrete and subjective criteria, and the results are difficult to compare from one study to another due to the lack of a common protocol. Using this without-*a priori*-method, we reduce these limitations, and as consequence, help in studying archaeological faunal remains in a more quantitative manner. In this study, we were able to analyze and interpret the response of a specific skeletal part (i.e., the mandible) according to its mechanical properties and their impact on its preservation. We concluded that the mandible shows a systematic dichotomy in its preservation, with a higher *ramus* fragmentation than that of the mandibular *corpus*. However, there are non-negligible variations within each taxon that must be considered in the differential preservation analysis of an archaeological site. Future bioarchaeological research on agouti and rice rats in the Lesser Antilles archipelago will benefit from this method to perform geometric morphometric studies of the spatio-temporal variation of the taxa.

**Supplementary Materials:** The following supporting information can be downloaded at: https://www.mdpi.com/article/10.3390/quat5010014/s1, Table S1.1: Number of archaeological studied mandibles per taxa; Table S1.2: Number of Oryzomyini from the South American continent studied; Table S2: Number of identified specimens (NISP) of Oryzomyini and *Dasyprocta*; Table S3: Description of the 13 landmarks digitalized; Table S4: Description of the most common fragmentation patterns for Oryzomyini and *Dasyprocta*; Table S5: Oryzomyini most frequent patterns tested according to the three counts.

**Author Contributions:** Conceptualization, A.E. and M.D.; methodology, A.E. and M.D.; formal analysis, M.D. and A.E.; investigation, M.D., A.E. and R.M.; writing—original draft preparation, M.D., A.E., S.G. and V.N.; writing—review and editing, M.D., A.E., S.G., V.N. and R.M.; funding acquisition, S.G. and A.E. All authors have read and agreed to the published version of the manuscript.

**Funding:** This work was supported by the European Research Council (ERC) under the European Union's Horizon 2020 research and innovation program (grant agreement No. 852573); the PEPS ECOMOB grant of the CNRS, INEE (project MIGGRANT) and the ATM Blanche grant of the MNHN (project PILORI).

**Institutional Review Board Statement:** The study did not require ethical approval.

**Informed Consent Statement:** Not applicable.

**Acknowledgments:** We would like to acknowledge the curators and staff of the different museums and institutions for kindly providing access to the archaeological collections under their care, Guadeloupe (including Saint-Martin) and Martinique Departments of Archaeology (Regional Office of Cultural Affairs, French Ministry of Culture), Antigua and Barbuda National Parks and the Barbuda Research Complex (BRC), and the Florida Museum of Natural History in Gainesville (FLMNH). Additionally, to the curators and staff who provide access to samples in biological collections of sigmodontine, including the FMNH, AMNH, NMNH, MLP, MNUFRJ, and MZUSP. We are grateful to Alexis Durocher and Anthony Herrel for their help in language editing and revision. We also warmly thank the anonymous reviewers who gave helpful comments to improve the manuscript.

**Conflicts of Interest:** The authors declare no conflict of interest.

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
