# Peer review of "Defining Fragmentation Patterns of Archaeological Bone Remains without Typologies: A Landmark-Based Approach on Rodent Mandibula"

_quaternary, doi:10.3390/quat5010014_

Round 1

Reviewer 1 Report

This original study proposes a standardized landmarks-based protocol for the description and quantification of mandibular fragmentation patterns that could be applied for future taphonomic and zooarchaeological research. Traditional bone fragmentation typologies have been described based on subjective observations, so this work contributes to reducing observer subjectivity and achieving universality.

This is a novel and well-structured work. The research design is appropriate, and the analysis methodology is adequate. I recommend the publication of this work. There are some minor spelling corrections along the manuscript. A revision of the English could be useful.

Author Response

Dear reviewer,

First of all, I would like to thank you for your interest in this study, which is very important to us.

We have taken into account your comments and spelling corrections. In parallel, we have also had the English revised by an English speaker as you suggested. I hope that this new version will suit you.

King regards,

Marine Durocher and co-authors.

King regards,

Marine Durocher and co-authors.

Reviewer 2 Report

Dear authors,

I think that the manuscript is very interesting and could be useful for studies of recent prehistoric and historic sites, where rodent mandible appears more complete. I have only few questions and suggestions concerning various things that I notice in the paper. Following you can find my questions and suggestions:

-This method could be applied to other groups of rodents, that have a different morphology, as Arvicolines?, that are the most abundant species in the Pleistocene and early Holocene of northern hemisphere.

-in line 99 you said fine meshes, but 1 and 2.5 mm are not fine. Fine are less than 0.8 mm. You can point only “with two meshes”

-In 4.1 I don’t understand very well one thing. You are mixing MNI with total of specimens? In this case how many mandibles do you have of each group, not in MNI, in Number of Identified Specimens?

-In 5.3 why you don’t take into account the possible accumulation of the smaller group that you have by other predators, like nocturnal or diurnal birds of prey or small carnivores? Are you sure that all the material has been accumulated by humans?. You have to discuss a little bit this point.

Yours Sincerely

Author Response

Dear reviewer,

We thank you for your comments and remarks which we have taken into account as I hope you will see in the attached manuscript.

I would also like to take this opportunity to respond in more detail to the questions and suggestions you have raised.

Concerning the application of this method to other groups of rodents with a different morphology: While working on rice rats and agoutis we realized that this method could be applied to rodents with different mandibes morphologies. So if these two rodents were used here as an example, we can perfectly envisage that this method can be applied to other taxa such as the Arvicolines, but also to taxa that do not belong to the Rodentia group, an application is in fact underway on dog mandibles and we also plan to apply it to 3D models.

Concerning the meshes we have modified the text as you suggested.

About the MNI or the number of specimens. In our case it turns out that the two are the same but to avoid misinterpretation we have changed all references to MNI to Number of identified specimens (NISP). 

In section 5.3 we added the sentence: "Indeed, these remains come from human occupation sites and butchery cuts and burn marks present on the bones suggest an anthropic accumulation" which allows us to explain why we chose to exclude the possibility of accumulation by predators.

Yours sincerely,

Marine Durocher and co-authors

Reviewer 3 Report

This paper presents a novel and meaningful approach to solve the problems of identifying fauna characterised by a high degree of fragmentation due to different taphonomic conditions. It presents a clear and relevant structure, the sample analysed is adequate for the proposed aims, especially in the case of rice rat. The lower proportion of agoutis is adequately justified. We make a positive assessment of the classification of the studied mandibular remains into three categories according to provenance, excavation characteristics and recovery procedures, because of their direct impact on the taphonomic processes affecting the remains.

The method is well explained and sufficiently developed. The graphical complement accompanying the text is relevant and contributes to its comprehension. The results obtained show several achievements, which we value positively:

Firstly, the standardised protocol based on landmarks for the description of mandibular fragmentation patterns with quantifiable and objective indicators, especially as it highlights a procedure applicable to other assemblages.

Secondly, it is important to have demonstrated that the preservation of the mandibles differs between the two taxa, since this suggests that they should be treated differently in future studies.

Likewise, the four aspects covered in the discussion are relevant to propose, without a doubt, the publication of this paper in this journal.

Author Response

Dear reviewer,

Thank you very much for your interest in this article and our research.

The method we present here seems important to share with the rest of the scientific community and we thank you for having deemed these results worthy of publication.

Yours sincerely,

Marine Durocher and co-authors